# A Simple Model for Halogen Bond Interaction Energies

**Robert A. Shaw** and **J. Grant Hill** *

Department of Chemistry, University of Sheffield, Sheffield S3 7HF, UK; rashaw1@sheffield.ac.uk
* Correspondence: grant.hill@sheffield.ac.uk; Tel.: +44-(0)-114-222-9392

**Abstract:** Halogen bonds are prevalent in many areas of chemistry, physics, and biology. We present a statistical model for the interaction energies of halogen-bonded systems at equilibrium based on high-accuracy ab initio benchmark calculations for a range of complexes. Remarkably, the resulting model requires only two fitted parameters, $X$ and $B$—one for each molecule—and optionally the equilibrium separation, $R_e$, between them, taking the simple form $E = XB/R_e^n$. For $n = 4$, it gives negligible root-mean-squared deviations of 0.14 and 0.28 kcal mol$^{-1}$ over separate fitting and validation data sets of 60 and 74 systems, respectively. The simple model is shown to outperform some of the best density functionals for non-covalent interactions, once parameters are available, at essentially zero computational cost. Additionally, we demonstrate how it can be transferred to completely new, much larger complexes and still achieve accuracy within 0.5 kcal mol$^{-1}$. Using a principal component analysis and symmetry-adapted perturbation theory, we further show how the model can be used to predict the physical nature of a halogen bond, providing an efficient way to gain insight into the behavior of halogen-bonded systems. This means that the model can be used to highlight cases where induction or dispersion significantly affect the underlying nature of the interaction.

**Keywords:** halogen bond; theoretical chemistry; intermolecular interactions

## 1. Introduction

Halogen bonds are an important class of non-covalent interaction where a halogen-containing donor, AX, interacts with a Lewis base as acceptor, B. While examples of halogen bonds were recognized as early as 1814 [1–3], it is only more recently with detailed X-ray diffraction [4–6] and spectroscopic [7–9] studies that they were found to be prevalent in both the gas and condensed phases [10,11]. These investigations discovered several striking properties, in particular the strong preference for linear geometries [12,13], where the AX···B angle is close to 180°, and interaction energies similar to those of hydrogen bonds [8,14]. These factors give halogen bonds a high degree of tuneability, making them ideal for use in fields ranging from crystal engineering to nanomaterials and drug design [11,15–23].

Halogens are conceptually seen as being electron rich, making their interaction with similarly electronegative bases counterintuitive. The most popular recent explanation is that of a $\sigma$-hole, first suggested in 2005 by Clark et al. [24]. They posit that the attachment of a suitably electron-withdrawing group to a halogen atom results in withdrawal of electron density from the halogen along the $\sigma$-bond. This withdrawal results in a charge anisotropy such that there is a positive cap on the face of the halogen atom opposite the bond. As such, a simple electrostatic argument can be made for how Lewis bases then interact with the halogen, and this explains the strong geometry dependence. It has been proposed that this is just an example of a wider class of analogous interactions, with similar effects seen for chalcogens, pnictogens, and tetrels [18,25–28]. The electrostatic potential can be calculated, and measured experimentally, confirming that the anisotropy does indeed exist [29–31]. It has been found that the $\sigma$-hole increases in size and intensity as one goes down the group, and that the strength

of the interaction increases with the electron-withdrawing power attached to the halogen [32–36]. These factors lend support to the intuitive explanation.

The above points clearly indicate that electrostatics are important in such systems. However, the "makeup" of halogen bonds has attracted considerable debate [37–47]. It has been comprehensively shown that electrostatics alone are not sufficient to fully describe these interactions [9,48]. The IUPAC definition of the halogen bond emphasizes that "the forces involved in the formation of the halogen bond are primarily electrostatic, but polarization, charge transfer, and dispersion contributions all play an important role" [49]. Indeed, in 1996 an analysis of the Cambridge Structural Database combined with intermolecular perturbation theory calculations came to the same conclusion as the IUPAC definition when considering interactions between carbon-bonded halogens and electronegative atoms [50]. As such, the $\sigma$-hole description, while conceptually very useful, is only part of the story. Numerous studies have demonstrated that dispersion is a very important component in differentiating halogen bonds from one another [12,44,51,52]. Similarly, exchange-repulsion effects can have a large impact on the geometries of halogen-bonded systems [12,44,48], and charge transfer is argued to be a distinguishing factor in many cases [39,45,46,53].

It has been pointed out that almost all such phenomena come under the umbrella of polarization [41]. Certainly, significant charge transfer such as found in so-called "Mulliken inner complexes" is often represented in the form $AX^- \cdots B^+$ [54], suggesting an extreme form of polarization. It is also true that dispersion, which is the purely mechanical interaction due to the instantaneous fluctuations of electrons, can be formally derived from polarizabilities. Exchange-repulsion is somewhat distinct but necessarily contaminates all other terms. This does not mean that such decompositions are meaningless, but rather that they should be treated with caution. It is sensible to distinguish "local" polarization and charge transfer, as this allows for a simple descriptor of when certain systems may behave substantially differently and uses an idea that is well-established within both the experimental and theoretical communities. The local distinction in this context simply refers to distortions and anisotropies in the electron density of a molecule constrained primarily to said molecule, as opposed to distortions effected by the surrounding environment resulting in substantial transfer of density away from the original molecule. Several examples of such unusually strong interactions have been reported [45,46,55–58], and there has been experimental evidence for charge transfer, from both rotational [12,59] and X-ray absorption [60] spectroscopy.

In a similar vein, the knowledge that dispersion is important implies that certain theoretical methods will not be useful. As this interaction is by definition due to the dynamical correlation of electrons, uncorrelated mean-field methods such as Hartree-Fock will not give accurate results. In particular, most density functionals are known to perform poorly on such systems [61–66]. The combination of this and the fact that non-covalent interactions involve small energy differences means that only the highest accuracy theoretical methods consistently give results in agreement with experiment. These are prohibitively expensive, however, and restricted to fairly small molecular complexes. The most important applications involve large, extended systems in the condensed phases, which are also difficult to study experimentally. As such, it is of considerable interest to find simple, reliable methods to accurately predict the strength of halogen bonds. Equally, quantitative measures for distinguishing when a system will behave substantially differently to other, similar examples could provide insight into the nature of these important interactions.

One interesting approach was used by Legon and Millen for hydrogen bonds [67]. They considered experimental spectroscopic force constants, $k_\sigma$, which are closely linked to the interaction energy, and parametrized the molecules involved to give a simple prediction of these force constants in new hydrogen-bonded systems. The model was

$$k_\sigma = cNE \qquad (1)$$

where $c$ is a proportionality constant, while $N$ and $E$ are termed the "nucleophilicity" and "electrophilicity" of the hydrogen-bond acceptor and donor, respectively. Despite their name,

no physical basis was suggested for these; they are empirically derived parameters found by comparing to values of one for a model system, in this case $H_2O\cdots HCl$. Tests of this model showed remarkably small deviations from experiment of less than 0.5 kcal mol$^{-1}$ in many cases, although unsurprisingly these errors increased upon extrapolation to new systems. More recent work has attempted to extend this approach to other types of non-covalent interaction, including halogen bonds [68,69]. Such a model would be ideal for halogen-bonded systems as it requires minimal effort. High-accuracy calculations or experiments would only be needed for a small number of "standard" systems before the parameters so determined could be used to quickly predict interaction strengths in new complexes.

In light of this, the present study has three aims. Firstly, high-accuracy benchmark calculations are presented for a wide range of small molecular systems. These data will then be used to investigate various simple models, demonstrating in Sections 2.1–2.3 that a similar approach to that in Equation (1) very accurately describes many halogen-bonded complexes. Perhaps most importantly, the theoretical basis for the analysis is investigated in Section 2.4, providing insight into the nature of halogen bonds and allowing for the development of criteria to distinguish substantially different subclasses of interaction. The approach is also tested on larger, more practically relevant systems, giving results that are at least as good as the best density functionals for non-covalent interactions.

## 2. Results and Discussion

To formulate a model for halogen bonds, systems of interest need to be selected and divided into two groups: fitting and validation sets. The criteria for which systems are to be considered are that they be tractable by the high-accuracy computational methods to be employed, and that they be representative of known halogen-bonded complexes. In practice, this restricts our pool of candidates to small molecules (less than 10 atoms) in the gas phase, many of which have been studied extensively using spectroscopy [7,70–74].

For the fitting set, the halogen-bond donors were all chosen to be diatomics of the form AX, where A = H, F, Cl, or Br, and X = Cl, Br, or I. These have a broad range of electrostatic properties, with for example electric dipole moments ranging from weakly negative (from X to A, e.g., for HBr) to very strong positive dipole moments, as in FI. The $\sigma$-hole model intuitively predicts that the size of the positive hole on the halogen acting as the halogen-bond donor should be larger the more positive this dipole moment, and that more polarizable atoms (such as iodine compared to bromine) will have larger holes. The halogen-bond acceptors (Lewis bases) were chosen to be $H_2O$, $CH_2O$, $H_2S$, $CH_2S$, HCN, and $H_3N$, covering the most commonly found acceptor atoms (O, N, and S) in different environments.

The validation set, on the other hand, was purposefully chosen to have a more diverse selection of systems. The halogen-bond donors were $F_2$, $Cl_2$, and $CF_3X$ where X = Cl, Br, or I; crucially, the latter three are no longer diatomics. Similarly, the acceptors were larger, comprising methanol, ethene, oxirane, thiirane, and phosphine. Of particular note is the inclusion of a $\pi$-to-halogen bond, and a different acceptor atom in phosphorous. Complete basis set (CBS) limit CCSD(T)-F12b counterpoise-corrected interaction energies and geometries for all systems can be found in the Supplementary Materials (SM). In agreement with previous investigations [12,75], the interaction energies are found to be sensitive to small changes in geometries, and to the size of the basis set. In particular, correctly identifying the extent to which the AX bond length increases on complex formation is vital in accurately determining the interaction strength. Notably the geometries agree well with spectroscopic data where available, and the predictive rules of Legon [76].

### 2.1. Model Fitting

From a statistical viewpoint, the two simplest models that could be suggested for the interaction energy, $E_{ij}$, between a halogen-bond donor with parameter $X_i$ and acceptor with parameter $B_j$ involve either a linear or product combination:

$$E_{ij} = X_i + B_j + c \tag{2}$$

$$E_{ij} = cX_iB_j \tag{3}$$

where $c$ is a real constant setting the energy scale, the latter being of the same form as Equation (1). However, we do not fit the parameters by arbitrarily choosing a single halogen-bond donor and acceptor to have unit parameters, as was done by Legon and Millen [67]; instead, we use an unbiased fitting over all molecules in the fitting set, as described earlier. These parameters are purely statistically fitted values, which can be found in the Supplementary Materials, and we ascribe them no specific physical meaning. A more physically motivated model might also include a distance dependence. Defining $R_{e,ij}$ to be the equilibrium separation between the donating halogen atom and the accepting atom on the base, a simple Coulombic model would suggest a dependence on $R_{e,ij}^{-1}$, whereas if the interaction were dispersive in nature, the classical dependence would be $R_{e,ij}^{-6}$. This could be included by modifying either of Equations (2) and (3) by multiplying by $R_{e,ij}^{-n}$ for some integer $n$, or by adding a weighted correction depending upon it. The former will be particularly important, and we define the *Pn* model as being that of the form

$$E_{ij}^{Pn} = \frac{cX_iB_j}{R_{e,ij}^n} \tag{4}$$

Thus, Equation (3) would be the *P*0 model. There are infinite other possibilities, including allowing for multiple parameters per molecule; this risks severely overfitting, however, given the fitting set only has approximately four points per molecule. The restriction of $n$ to integer values is motivated by analogy to standard expressions for the potential energy of interactions between stationary multipoles; however, as discussed below, the removal of this restriction to then allow non-integer values of $n$ would have little impact in the performance of the model. We should stress at this point that we are categorically not suggesting that these models describe a geometric *dependence* of the energy. Rather it is the total interaction energy *at equilibrium* that is being described, with the strength mediated by the intermolecular separation. However, it is also incorrect to say that this is entirely independent of any physical dependence of the energy on the separation: as the parameters are fitted across a set of molecules, the dependence on $R_e$ cannot simply be absorbed into the parameters $X$ and $B$, and must represent an independent factor in the model.

An important indicator of the validity of a fitting procedure is the distribution of the residuals, or equivalently, the correlation between the predicted and actual values. In Figure 1, the predicted vs. actual energies are plotted, demonstrating that the *P*0 model is a much better fit than Equation (2). Crucially, it appears to abide by the assumptions of the fitting procedure, namely the assumption of normality of errors. This was not the case for the linear model, or any model with an added (rather than multiplied) $R_e^{-n}$ correction term. The fitted parameters under both models can be found in the Supplementary Materials, with some further discussion of the *P*4 parameters below. Summary statistics for several different models are given in Table 1, showing that the product models have by far the lowest errors. Both the *P*0 and *P*4 models have low root-mean-square errors (RMSEs), their maximum errors are less than 1 kcal mol$^{-1}$, and a mean-signed error close to zero indicates there is no systematic under- or over-estimating of the interaction energy. On the other hand, it is clear that a simple weighted dispersion model, i.e., a $E_{ij} = kR_{e,ij}^{-6}$ model where $k$ is optimized as a parameter fixed across all molecules, does not perform well. The statistics presented in Table 1 are for the fitting set of complexes, that is, the same complexes that the parameters were fit to, hence good performance is somewhat expected. The quality of the interaction energies predicted by the product models for complexes not included in the fitting set is examined below.

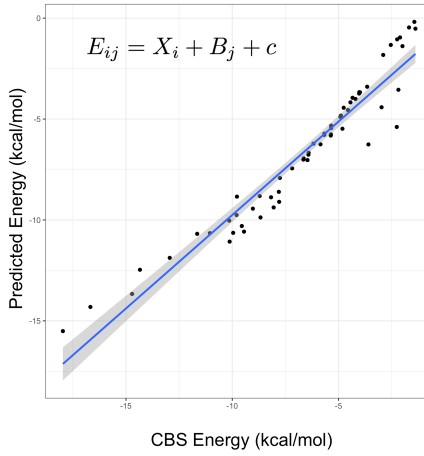
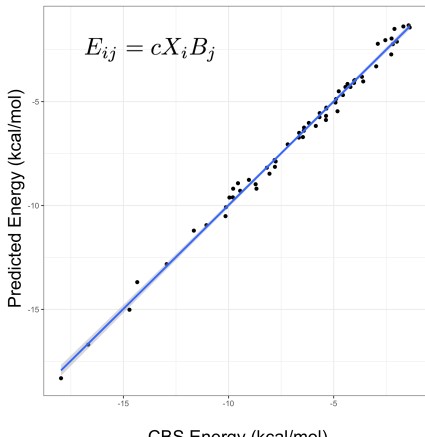

**Figure 1.** The predicted versus true interaction energies for the linear (**left**, Equation (2)) and *P*0 (**right**, Equation (3)) models. In the former, a non-linear trend is seen, suggesting non-normality of errors. The gradient and adjusted $R^2$ value of the line in the right-hand figure are 1.0 and 0.995, respectively. A perfect model would have unit gradient and zero intercept.

**Table 1.** Summary statistics for the linear, *P*0, $kR_{e,ij}^{-6}$, and *P*4 models over the fitting set of 60 complexes. These include the root-mean-square, maximum, mean-signed, and mean-absolute errors in kcal mol$^{-1}$.

| Model | RMSE | Max. | MSE | MAE |
|---|---|---|---|---|
| Linear | 1.13 | 3.13 | 0.00 | 0.75 |
| *P*0 | 0.30 | 0.68 | −0.01 | 0.24 |
| $kR_e^{-6}$ | 2.99 | 7.11 | 0.50 | 2.32 |
| *P*4 | 0.14 | 0.41 | 0.00 | 0.11 |

In addition to the accuracy of the predicted interaction energies, it is also possible to evaluate the models in terms of their efficient use of information, as quantified by the Akaike information criterion (AIC) [77]. This statistic roughly equates to whether the increase in complexity prescribed by adding the parameters is justified in relation to the amount of data supplied; a small number indicates the model is 'efficient' in its use of data. The AIC for the linear, *P*0 and *P*4 models is 204, 35, and 14, respectively, demonstrating that product models are more efficient than linear, and further justifying the increase in complexity of the added distance dependence in the *P*4 model.

*2.2. Principal Component Analysis*

The significance of the *P*0 model can be understood in its relation to a principal component analysis, a widely used technique for dimensionality reduction. In this case, if **E** is an $M \times N$ matrix of interaction energies $E_{ij}$, then a principal component analysis takes the form of a singular value decomposition:

$$\mathbf{E} = \mathbf{u}\Lambda\mathbf{v}^T$$

where $\Lambda$ is a diagonal matrix of $N$ singular values (or "components") $\lambda_i$, while **u** and **v** are $M \times N$ and $N \times N$ matrices of component vectors. In this way, any element of **E** can be written as

$$E_{ij} = \sum_{k=1}^{N} \lambda_k u_{ik} v_{kj} \tag{5}$$

If the principal component, $\lambda_1$, is much greater than all the other components, then we see that the sum in Equation (5) simply reduces to the *P*0 model in Equation (3), where $c = \lambda_1$, $X_i = u_{i1}$, and $B_j = v_{1j}$.

Performing this analysis on the matrix of interaction energies gives the principal component as being roughly 30 times larger than the second component, explaining 99.3 percent of the variance in the energies. A further 0.6 percent is explained by including the second component, with all further components being negligible. This explains both why the simple product model is strikingly successful, and a potential way to improve it by adding a second component in Equation (5). Moreover, it provides an easy way to parametrize $Pn$ models with $n > 0$, by forming a matrix with values $E_{ij}R_{e,ij}^n$ and performing a singular value decomposition. Figure 2 shows how the root-mean-squared error for the fitting set varies with $n$. Clearly, $n = 4$ provides the best results, and as can be seen in Table 1, is a substantial improvement on the $P0$ model, achieving accuracy beyond what can be achieved by using density-functional theory, as will be discussed shortly. As the $P4$ model has an RMSE of only $0.14$ kcal mol$^{-1}$ and Figure 2 demonstrates that the relationship between the error and the value of $n$ is clearly discontinuous, attempting to include non-integer values of $n$ would not substantially improve the model and would sacrifice simplicity, hence it has not been pursued.

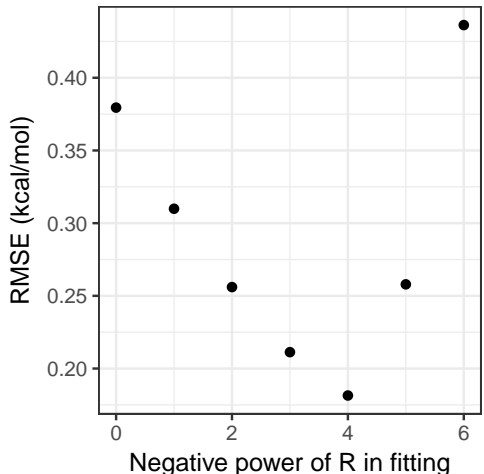

**Figure 2.** Root-mean-square error over the fitting set for the $Pn$ models, as a function of $n$.

While including a distance dependence clearly improves accuracy, it also introduces complications. The most obvious of these is the requirement for an estimate for the separation to be available. For many applications where this simple model would be useful, an estimate of the separation is readily available from, for example, rotational spectroscopy, X-ray crystallography, or computational chemistry calculations at low levels of theory. We note that a small error in $R_e$ of $\delta$ percent that may arise in such estimates can easily be shown to give an error of roughly $4\delta$ percent in the predicted interaction energy from the $P4$ model, which remains small. Moreover, as can be seen from Figures 1 and 2, the $P0$ model still performs very well with an RMSE of $0.30$ kcal mol$^{-1}$, and so could be used when no value for $R_e$ was available. Again, we also stress that the model has been developed for predicting the interaction energy at equilibrium; the functional form in Equation (4) will diverge to negative infinity at short distances. Similarly, only $n = 6$ would correctly recover the expected long-range behavior. Developing a model applicable to a range of displacements, somewhat akin to a Lennard-Jones potential, is beyond the scope of the current study.

A small sample of the parameters fitted to the $P4$ model are shown in Table 2, with a full set of parameters for all molecules, and for the linear and $P0$ models, provided in the Supplementary Materials. Focusing momentarily on the $X_i$ parameters for the halogen-bond donors, it can be seen that as the difference in electronegativity of the two halogen atoms increases, the value of $X_i$ also increases, increasing the likelihood of a strong halogen-bond. In the case of $F_2$ the value of $X_i$ becomes very small, which is consistent with $F_2$ forming weakly bound complexes with Lewis bases that arguably do not meet the established criteria for a halogen bond [35,49]. The $B_j$ parameters associated with the Lewis bases do not display any obvious trends; while harder bases (such as $H_2O$) tend to have $B_j$ values that

are smaller in magnitude than softer bases (such as $H_2S$), there is no correlation between $B_j$ and the absolute hardness of Pearson [78,79] when the full set of Lewis bases considered is examined. It is perhaps unsurprising that we have been unable to find a correlation between the model parameters ($X_i$ and $B_j$) and properties of the isolated monomers—the model has been fit to interacting complexes where a degree of polarization/perturbation of the charge distribution of a given monomer by its halogen bonding counterpart has taken place.

**Table 2.** Selected parameters for halogen-bond donors and Lewis bases as fitted to the *P*4 model. The optimized value of *c* for this model is 3327.9474 kcal mol$^{-1}$ Å$^4$, all other parameters are dimensionless. A table of all parameters can be found in the Supplementary Materials.

| Halogen-Bond Donor | $X_i$ | Lewis Base | $B_j$ |
|:---:|:---:|:---:|:---:|
| $F_2$ | 0.0621 | $H_2S$ | −0.4643 |
| FCl | 0.2215 | $CH_2O$ | −0.3056 |
| FBr | 0.3306 | $H_3N$ | −0.4416 |
| FI | 0.4600 | $H_2O$ | −0.2947 |

*2.3. Validation and Comparison with Other Methods*

The validation set comprises the five new halogen-bond donors described above paired with all six original acceptors, and the five new acceptors paired with the ten original donors. As such it constitutes a larger set (80 systems, as opposed to 60 in the fitting set). However, during our investigations it became apparent that some of the systems behave markedly differently to any of the others. Specifically, those involving FCl, FBr, and FI interacting with phosphine and thiirane. These exceptional cases have been discussed elsewhere [46,55], and were excluded from the validation set as their errors across all methods were over an order of magnitude larger than for any other systems. This was true also for the density functionals considered, both of which significantly under-bound the complexes, by as much as 8 kcal mol$^{-1}$ in the case of FCl$\cdots$PH$_3$.

The relevant $X_i$ and $B_j$ parameters for all new molecules were found by calculating the CBS limit CCSD(T)-F12b energy for the interaction with water for the halogen-bond donors and of the acceptors with BrI. These energies were then divided through by the known parameter ($X_i$ or $B_j$) and the calculated $R_{e,ij}^{-n}$. This unnaturally results in ten systems with zero error, and as such these data were excluded from the subsequent error analysis. The bond lengths in the model for the remaining systems were taken to be those calculated using M06-2X, to give a fair and consistent comparison with later results where CCSD(T) level calculations are computationally intractable.

The error distributions for the *P*4 model over both the fitting and validation sets, along with those calculated using the M06-2X and $\omega$B97X-D density functionals at the aVTZ level, are shown in Figure 3, with the equivalent plot for the *P*0 model given in the Supplementary Materials. These functionals were chosen as previous benchmarks have shown them to be particularly good for halogen bonding interactions [75]. From the Figure, however, we see that the simple product model is performing at least as well, if not better than, these functionals. In particular, both the fitting and validation data are centered around zero residual error, indicating no systematic bias, which contrasts with the two functionals, which systematically over- and underestimate the energies slightly, for M06-2X and $\omega$B97X-D respectively. Moreover, the overall spread is narrow, and mostly concentrated around zero, staying consistently within nominal "chemical accuracy" of 1 kcal mol$^{-1}$. This is opposed to M06-2X, which shows a much more protruded density, significantly overpredicting some energies.

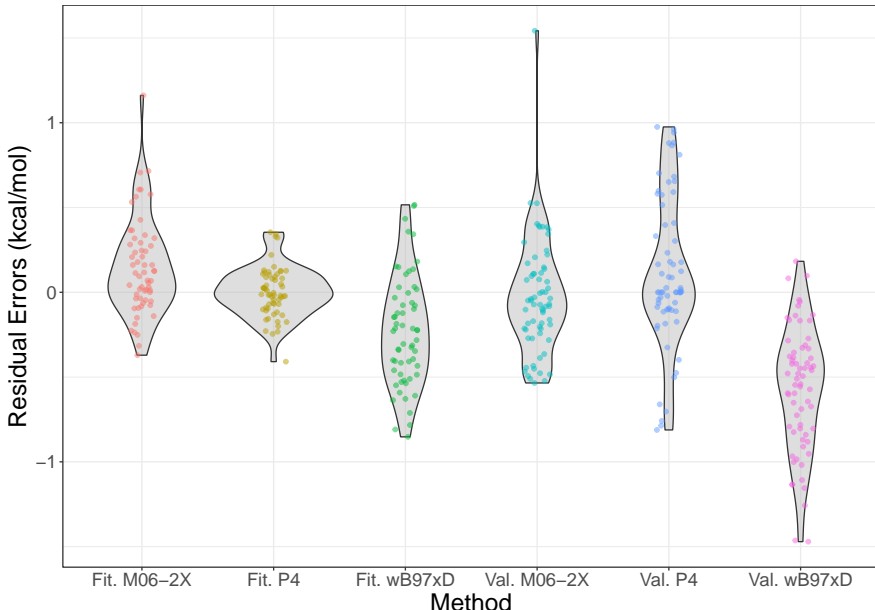

**Figure 3.** Violin plots of the error distributions of the *P4* model, M06-2X/aVTZ, and $\omega$B97X-D/aVTZ, compared to CCSD(T)-F12b/CBS results. The model is split into data from the fitting (Fit.) and validation (Val.) sets. The shape of the violin shows where the density of errors is concentrated—i.e., the frequency with which errors are found in a small interval—such that an ideal distribution would be a very short, wide density centered on the origin. Please note that the density is plotted symmetrically about the vertical axis, and the horizontal scale is relative (so it is the same for all the violins); the total area of a violin integrates to the number of points, the width representing a proportion of the total number. The individual data points have also been plotted, with a small amount of jitter added in the horizontal direction to aid visibility.

The mean-absolute errors for the validation set with the *P4* model, M06-2X and $\omega$B97X-D are 0.28, 0.36, and 0.30 kcal mol$^{-1}$. These are all broadly similar, but it should be noted that a combination of Shapiro-Wilk and Kolmogorov-Smirnov tests indicate that the error distributions for each are normally distributed [77], but drawn from distinct distributions, with $p < 0.01$ in each pairwise comparison. For reference, MP2/aVTZ results gave an MAE of 0.77 kcal mol$^{-1}$, almost three times that of the product model.

The performance of the statistical model is astonishing, as it gives better accuracy than high-level quantum chemical methods at a fraction of the cost. For any new complex of interest, the relevant parameters can be determined from a single calculation with a reference molecule (water or BrI), and then reused in all other contexts.

To test this, calculations were performed on considerably larger molecules than those in the fitting or validation set, where using the high-level coupled cluster method would be unfeasible. Based on M06-2X producing an error distribution that is much more centered around zero for the validation set than $\omega$B97X-D (see Figure 3), M06-2X/aVTZ calculations were performed for the halogen-bond acceptors sulphoximine, glycine, valine, and leucine, and the donors $C_6F_5X$ with X = Cl, Br, and I. The interacting atom on the acceptors were the nitrogen in sulphoximine and the carbonyl oxygen on the amino acids; geometries can be found in the Supplementary Materials. The parameters for the model were determined with respect to the reference molecules at the same level of theory. Table 3 shows the results of these tests. Despite being extrapolated to calculations with different systems, not involving any of the original fitting data, the mean-absolute deviation is 0.49 kcal mol$^{-1}$. The mean-absolute deviation of the *P4* model from the M06-2X results across the fitting and validation sets is 0.48 kcal mol$^{-1}$, suggesting that similar error levels have been maintained despite the

significant increase in molecule size. The *P*4 model therefore represents a rapid and accurate approach to predicting the interaction energies of halogen-bonded systems.

**Table 3.** The energies for each pair of new halogen-bond acceptor and donor at the M06-2X/aVTZ level, along with the energies predicted by the model, in kcal/mol.

| Lewis Base | $C_6F_5Cl$ | | $C_6F_5Br$ | | $C_6F_5I$ | |
|---|---|---|---|---|---|---|
| | **M06** | **Pred.** | **M06** | **Pred.** | **M06** | **Pred.** |
| Sulphox. | −3.32 | −4.06 | −4.61 | −4.95 | −5.96 | −6.48 |
| Glycine | −2.49 | −3.14 | −3.66 | −3.83 | −5.18 | −5.01 |
| Valine | −3.75 | −3.68 | −4.58 | −4.49 | −6.22 | −5.87 |
| Leucine | −5.12 | −4.04 | −4.97 | −4.94 | −6.45 | −6.46 |

The validation of the *P*4 model demonstrates that the $X_i$ and $B_j$ parameters are transferable to complexes outside of the original training set, indicating that a single parameter for a given monomer can be used in the prediction of equilibrium interaction energies of halogen bonds that presumably have a somewhat different underlying nature in terms of intermolecular forces. As detailed in the Introduction, the IUPAC definition of a halogen bond states that polarization, charge transfer and dispersion all play a role in this interaction, and it is well-known that the exact composition of these forces give rise to different strengths of interaction. In the next section we investigate the underlying nature of several halogen bonds and rationalize how the simple model results in transferable parameters for interactions with varying forces evident in the decomposition of the interaction energies.

## 2.4. The Nature of the Halogen Bond

The principal component analysis has allowed for greater insight into the mechanics behind the product model, and for elucidation of the distance dependence of the interactions. It also suggests that a method to improve the performance of the model further would be to include the second component in Equation (5), which leads to an expression of the form:

$$E_{ij} \approx cX_iB_j + d\chi_i\beta_j$$

where *d* is the second component, and $\chi_i$ and $\beta_j$ are second parameters for the halogen-bond donor and Lewis base, respectively. This use of a second component is impractical for two main reasons: it would double the number of parameters, which we have seen leads to severe overfitting; it would complicate the determination of new parameters, as two reference calculations would be needed, and a system of linear equations would have to be solved. However, the instances where the second component is important could serve as an indicator as to which systems behave differently to the norm and second component parameters fit to the *P*0 model are provided in the Supplementary Materials.

To this end, symmetry-adapted perturbation theory (SAPT) calculations were carried out, providing a decomposition of the interaction energies in terms of the physically relevant quantities of electrostatics, exchange, induction, and dispersion. Charge transfer can be separated out from the induction energy [53,80], as has been seen to be important for the phosphine systems [46], but we do not do that here as the best approach to doing so is not clear. The energy contributions for each system in the fitting set are given in the Supplementary Materials, along with figures illustrating each component as a percentage of the total interaction. Figure 4 shows how the percentage error in predictions from the *P*4 model compares with the relative importance of induction and dispersion in the SAPT decomposition of the interaction energy, with the results split by halogen-bond donor to show how trends in each quantity correlate. The induction and dispersion terms are both presented as ratios relative to the SAPT electrostatic term. It is immediately apparent that complexes where the model displays the largest percentage errors relative to the CCSD(T)-F12b/CBS data (Figure 4a) are those with significantly

increased relative dispersion (Figure 4b) or induction (Figure 4c) contributions. An increase in induction (potentially charge transfer) also appears to be concomitant with a decrease in dispersion, and vice-versa. Perhaps most interestingly, it is the halogen-bond donors HBr and HI that show the largest percentage errors, and consequently the largest proportion of dispersion along with the smallest proportion of induction. This suggests that these interactions are predominantly dispersive rather than electrostatic, in line with what we would intuitively expect given the relative electronegativities of hydrogen and the halogen atoms.

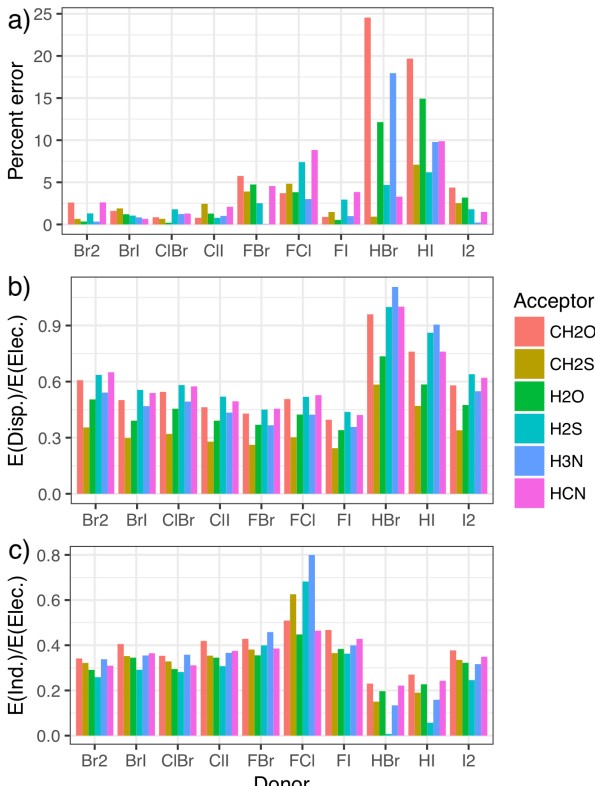

**Figure 4.** The error [relative to the CCSD(T)-F12b/CBS values] as a percentage of the overall interaction energy for the *P*4 model (**a**) compared with the ratio of the dispersion (**b**) and induction (**c**) contributions to the electrostatic component of the symmetry-adapted perturbation theory of the energy.

Additionally, the induction contribution shown in Figure 4c noticeably increases for the F–*X* donors, peaking for F–Cl. This agrees with trends noted for substantial charge transfer, namely the switching of the mode of binding to a Mulliken inner complex. This is again accompanied by a decrease in dispersion, and a pronounced increase in the errors from the simple *P*4 model. In both cases, the inclusion of the second component in the model almost entirely corrects for these differences, as can be seen in Figure 5. The second component reduces the strength of the interaction in situations where there is a large induction contribution (such as FCl in the top left), and increases the strength for those with a small proportion of induction (HI and HBr in the bottom right). Recalling that Figure 4 shows that a decrease in induction correlates with an increase in dispersion, this indicates that the single component *P*4 model underestimates the strength of interactions with a large dispersion contribution.

Moreover, it suggests that far from induction and dispersion being unimportant for the other systems, it is more that when combined they are of similar enough magnitude to one another that these effects are included in the fitting process. Inclusion of the second component in the model reduces the RMSE of the fitting data from 0.14 to 0.02 kcal mol$^{-1}$, a modest dimensionality reduction from six components to two. For practical use of the model, inclusion of these effects is irrelevant. The significance comes

from the utility of deviation from the model in categorizing the physical nature of the halogen bond. In particular, whether that deviation is an under- or overestimation of the interaction, or equivalently the importance of the second component, indicates when a complex has changed from a "typical" halogen bond, where electrostatics is seen to dominate, to one that is dispersive or induction (possibly charge transfer)-based, respectively. It thus has the potential to provide insight with minimal effort.

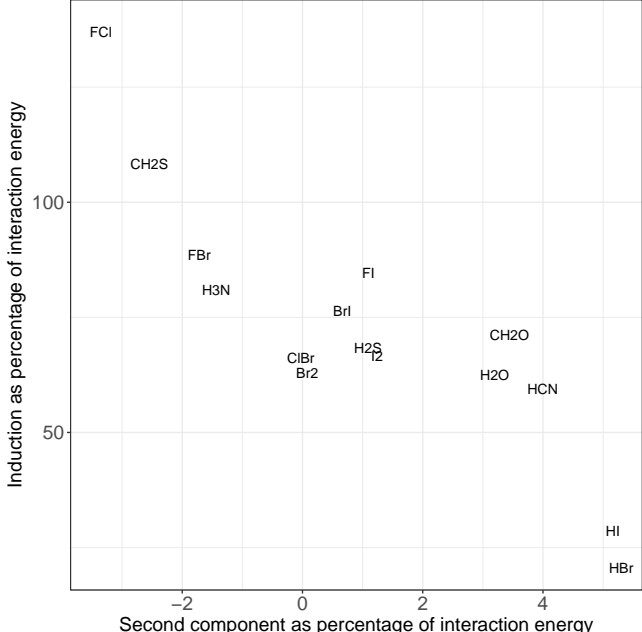

**Figure 5.** Comparison of induction energy with the energy due to the second component in the principal component analysis, each as a percentage of the total interaction energy, averaged over all systems containing the given molecule in the fitting set. Most systems fall in the middle, but those with larger dispersion (bottom right) or induction (top left) show a marked increase in the importance of the second component to the predicted energy. Please note that the values for ClI overlap those of BrI, so we only show the latter for clarity.

To further demonstrate the simple model halogen bond and how it can be used, we have prepared an interactive Jupyter notebook that is available via GitHub [81]. This includes a walkthrough of a simplified version of the fitting and analysis of the model, and an example of how parameters for new halogen-bond donors and acceptors can be found (potentially requiring only a DFT interaction energy calculation with a previously parameterized monomer and dividing out the known quantities in Equation (4)). In addition to acting as an explanation of the analysis in the present investigation, it is intended that the Jupyter notebook could also act as template for attempting to find a simple statistical model for other types of $\sigma$-hole-based interactions, such as chalcogen bonds.

## 3. Materials and Methods

Explicitly correlated coupled cluster calculations with singles, doubles, and perturbative triples [82] were carried out in the MOLPRO suite of programs [83,84] using the 3C(Fix) ansatz and approximation b, [CCSD(T)-F12b] [82,85,86], with a geminal Slater exponent of 1.0 $a_0^{-1}$. The cc-pV$n$Z-F12 basis sets were used, with the exception of Br and I, which used the cc-pV$n$Z-PP-F12 sets with the Stuttgart-Cologne small-core relativistic pseudopotential [87–90]; we note that no discontinuities are seen in trends going from chlorine to bromine when the pseudopotentials are introduced. Although not apparent from the abbreviation, these basis sets include augmentation with diffuse s and p functions. Geometries were optimized at the $n = $ T level, while single-point energies were calculated for $n = $ T, Q, then used to extrapolate to the CBS limit using the method described by Hill and coworkers [91]. The Fock and exchange matrices were density fitted using the cc-pVQZ/JKFit auxiliary basis for all atoms other than

bromine and iodine, which used the def2-QZVPP/JKFit sets [92,93]. All subsequent two-electron integrals were fitted using the aug-cc-pVQZ and cc-pV$n$Z-PP-F12 MP2Fit sets for the lighter and post-d elements, respectively [88,94]. The CABS+ procedure was carried out using the auxiliary sets specifically matched to the orbital basis [82,88,95–98] and the CABS singles correction was applied to the Hartree-Fock reference energy. The full counterpoise correction of Boys and Bernardi was used for all interaction energies [99].

Calculations with the M06-2X [100] and $\omega$B97X-D [101] density functionals were performed in Gaussian 09 [102], with the UltraFine integration grid. These functionals have been shown to perform particularly well for non-covalent interactions [75]. Symmetry-adapted perturbation theory [103] calculations at the SAPT2+(3)$\delta$MP2 truncation [104,105] were carried out in the SAPT2012 program [106] interfaced to MOLPRO, with the so-called "chemist's grouping" [107]. The aug-cc-pV(T+d)Z basis sets (abbreviated here as aVTZ) were used in both cases [108–110].

All errors quoted in calculated values are deviations relative to the CCSD(T)-F12b/CBS limit value, which has previously been shown to closely follow the same trends as experimental intermolecular force constants [47]. The errors are assumed to be normally distributed in any statistical analyses. As such, models were fitted to the data by minimizing an ordinary least-squares loss function of the errors using a quasi-Newton-Raphson procedure. The variable step size of Snoek et al. was applied [111], as well as Tikhonov regularization.

## 4. Conclusions

We have presented a statistical model for the interaction energy of halogen-bonded systems at equilibrium that takes the simple form $X_i B_j / R_{e,ij}^4$ (denoted $P4$), where $X_i$ and $B_j$ are parameters for the halogen-bond donor and acceptor, while $R_{e,ij}$ is the equilibrium separation between the two molecules. Using a regularized least-squares regression this model was fitted to benchmark quality data from the high-accuracy CCSD(T)-F12b method extrapolated to the CBS limit, for a set of 60 halogen-bonded complexes. Various alternative models were tested, but product models gave the best results. The mean-absolute and maximum errors in the calculated halogen-bond interaction energy over the fitting set for $P4$ were 0.11 and 0.41 kcal mol$^{-1}$, respectively. This represents greater accuracy than the M06-2X and $\omega$B97X-D density functionals, and is also the case when extended to 74 validation systems not in the original fitting set. Most promisingly, when extended to much larger and completely new complexes using a method (M06-2X) that is much less expensive than CCSD(T)-F12b, accuracy was maintained relative to the density-functional theory calculation, achieving root-mean-square deviations of less than half a kilocalorie per mole. A simpler version of the model of the form $E = X_i B_j$ ($P0$) also performs well, with mean and maximum errors for the fitting set of 0.24 and 0.68 kcal mol$^{-1}$, respectively. This reduction in accuracy is offset by convenience as the simpler model does not require knowledge of the equilibrium separation. The ease of parametrization and speed of prediction inherent to using either product model makes them potentially very useful for the rapid evaluation of interactions in, for example, virtual screening-like applications of a library of supramolecular synthons. Interactions predicted to be within a specific strength range can be quickly identified and the appropriate molecules proposed for computationally expensive calculations on large supramolecular systems.

The performance of the $P4$ model for the validation set indicates that a single parameter per donor or acceptor is transferable across a range of halogen bonding interactions, each with a different composition of underlying intermolecular forces. A SAPT analysis demonstrated that some of this transferability is due to the induction and dispersion components of the interaction energies summing to a similar magnitude in the majority of cases, hence the effects are included in the fitting process. Principal component analysis found that cases where a second component (adding a second set of parameters to the model) became substantial were found to correlate with increases in dispersion or induction contributions to the energy. These correspond to under- and overestimation of the interaction energy by the principal component, respectively, and thus provide an indicator for major changes in the underlying physical nature of the halogen bond, such as Mulliken inner complexes. While it is not

possible to tell *a priori* whether the second component will be important, the one-component models work sufficiently well for most systems, such that their failure could be used as an indicator. As $\sigma$-holes have been identified as playing a role in intermolecular interactions involving other p-block elements, including chalcogens, pnictogens, and tetrels, it is plausible that the applicability of the simple model is not restricted to halogen bonds. The current approach could easily be applied, perhaps elucidating both similarities and differences between many classes of non-covalent interaction. Particularly interesting would be to extend the analysis to off-equilibrium geometries, potentially leading to a simple model for the geometric dependence of the interaction strength. The emphasis here is on the simplicity of the approach, allowing back-of-the-envelope type calculations with the accuracy of computationally expensive quantum chemical calculations.

**Supplementary Materials:** The following are available online at http://www.mdpi.com/2304-6740/7/2/19/s1, model parameters for all molecules; benchmark energies and separations for the fitting and validation sets; DFT results for all systems; additional figures; Cartesian coordinates of benchmark geometries.

**Author Contributions:** Conceptualization, J.G.H.; methodology, R.A.S. and J.G.H.; validation, R.A.S. and J.G.H.; formal analysis, R.A.S. and J.G.H.; investigation, R.A.S.; writing—original draft preparation, R.A.S.; writing—review and editing, J.G.H.; visualization, R.A.S.; supervision, J.G.H.

**Funding:** This research received no external funding.

**Acknowledgments:** The authors thank Fred Manby for helpful conversations and Lee Brammer for comments on an earlier draft of the manuscript.

**Conflicts of Interest:** The authors declare no conflict of interest.

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
