# Peer review of "A Simple Model for Halogen Bond Interaction Energies"

_inorganics, doi:10.3390/inorganics7020019_

Reviewer 1 Report

The research work is well done and well written. I suggest acceptance of this paper for publication Inorganics.

Author Response

We thank the referee for their supportive comments on our work. There were no suggestions for improvement for us to address.

Reviewer 2 Report

A simple model for halogen bond interaction energies.

By Robert A. Shaw  and J. Grant Hill

The authors of have developed a simple statistical model for calculating halogen bond energies with relatively high accuracy. It is notable that the model (E = XB/Rne) requires only two fitted parameters (X for the donor and B for the acceptor) and optionally the equilibrium separation (Re). The root-mean-squared deviations (0.14 and 0.28 kcal/mol) are very small over a high number of separate fitting and validation data sets, respectively. The approach is particularly seducing and this simple model shows a high or even higher performance compared to the best density functionals generally used for non-covalent interactions at almost no computational cost. The authors demonstrate that the model can be applied to bigger molecules without a significant loss of accuracy. It can further be used to predict halogen bonding energies and identify cases which are affected by induction or dispersion interactions. The statistical approach is convincing and its simplicity, accuracy and rapidity makes the developed model very useful for the rapid evaluation and prediction of halogen bonding (and potentially also other sigma hole bonding interactions). The possible extension of the analysis to off-equilibrium geometries is an interesting perspective for reactivity issues.

Some minor corrections:

Line 151:  “at equilibrium” instead of “at equlibrium"

Line 351:  “3C(Fix) ansatz” instead of “3C(Fix) ansätz"

Figure 5:  should be improved for a better readability

Author Response

Point 1: Line 151:  “at equilibrium” instead of “at equlibrium"

This has been corrected.

Point 2: Line 351:  “3C(Fix) ansatz” instead of “3C(Fix) ansätz"

This has been corrected.

Point 3: Figure 5:  should be improved for a better readability

The data points / labels for BrI and ClI did overlap in this figure, which we agree makes it difficult to read. We have remedied this by removing the ClI data from the plot and adding the following text to the caption "Note that the values for ClI overlap with those of BrI, so we only show the latter for clarity".

Reviewer 3 Report

The authors describe in this interesting paper a statistical model for the interaction energies of halogen-bonded systems at equilibrium for a range of complexes

which can be used in much larger complexes whit good accuracy within 0.5 kcal/mol.

The paper is well written, and the referenced are adequate and overall the conclusions are fully supported by the result obtained from the performed experiments and therefore I recommend this paper for publication in Inorganics in the present form.

Author Response

We thank the referee for supporting publication in the present form.

Reviewer 4 Report

The article presents a simple model for interaction energy for halogen bonds. The authors used non-standard methods, I would say that even quite sophisticated. The authors focused not only on halogen binding energy but also presented additional aspects related to fitting and validation, or decomposition of energy according to SAPT. I have no objections, the article is written in a good language and very accessible.

The only thing I would like to read in this article is for which systems can a simple P0 or P4 model be used, and for which the second component in equation 5 should be included. This is not clearly specified. 

Perhaps such a short summary could be found in the conclusions.

Author Response

Point 1: The only thing I would like to read in this article is for which systems can a simple P0 or P4 model be used, and for which the second component in equation 5 should be included. This is not clearly specified. Perhaps such a short summary could be found in the conclusions.

The reviewer has picked up on an important point here, in that it is not possible to tell, without running the calculations, when the second component will be significant and warrant inclusion. In fact, it is when the P4 (or P0) model fails that perhaps we have the most interesting complexes. We have added the following text to the conclusions to summarise this "While it is not possible to tell a priori whether the second component will be important, the one-component models work sufficiently well for most systems, such that their failure could be used as an indicator."